# Symmetrization of Loss Functions for Robust Training of Neural Networks in the Presence of Noisy Labels and the Multi-class Unhinged Loss Function

## Abstract

Labeling a training set is not only often expensive but also susceptible to errors. Consequently, the development of robust loss functions to label noise has emerged as a problem of great importance. The symmetry condition leads to theoretical guarantees for robustness to such noise. In this work, we investigate a symmetrization method that follows from the unique decomposition of any multi-class loss function into a sum of a symmetric loss function and a class-insensitive term. We describe how this approach is related to regularization from Dirichlet priors on the outputs of the network. Notably, the special case of the symmetrization of the cross-entropy loss leads to a multi-class extension to the unhinged loss function. This loss function is linear but contrary to the binary case, it must have specific coefficients in order to satisfy the symmetry condition. Under appropriate assumptions, we show that this multi-class unhinged loss function is the unique convex multi-class symmetric loss function. It plays an interesting role among multi-class symmetric loss functions since the linear approximation of any symmetric loss function around points with equal components must be equivalent to the multi-class unhinged loss function. Remarkably, even though the cross-entropy loss is not inherently robust, it also exhibits this property. This means that around points assigning equal probabilities to every class, the cross-entropy will be approximately symmetric. Experiments on CIFAR10 validate the robustness of our approach.

## 1 Introduction

In recent years, deep learning has made significant advancements, achieving state-of-the-art performance in various domains such as computer vision and natural language processing (LeCun et al., 2015). However, these deep learning models often require extensive training on large datasets. Acquiring correct labels for such datasets can be costly. To mitigate this problem, crowdsourcing platforms have been employed, but they come with the drawback of potentially introducing high amount of errors into the labels. Zhang et al. (2017) tried to fit random labels on CIFAR10 and ImageNet with different deep neural network architectures. They came to the conclusion that deep networks can easily fit random labels during training and that their effective capacity is sufficient for memorizing the entire data set. This memorization ability can become particularly problematic in the presence of label noise, as the network may learn to fit the noise rather than the true underlying patterns. In order to address this problem, robust loss functions based on the symmetry condition have been proposed (Ghosh et al., 2015), Ghosh et al. (2017). For example, under the symmetry condition, the optimizers of the expected loss on the clean distribution are the same as the optimizers of the expected loss on the corrupted distribution with uniform label noise. The study and the design of new symmetric loss functions is therefore of great practical importance.

This work proposes a principled symmetrization method for multi-class loss functions leading to a general method for producing symmetric loss functions from non-symmetric loss functions. Our main contributions are the following:

i) We show that the proposed symmetrization method is related to a form of regularization with data-dependent Dirichlet priors (section 5).

ii) We apply the proposed symmetrization method to different loss functions (section 6). Notably, the symmetrization of the cross-entropy loss (CE) leads to a multi-class extension to the unhinged loss function (van Rooyen et al., 2015). Moreover, the symmetrization of the generalized cross-entropy (GCE) (Zhang & Sabuncu, 2018) gives rise to a loss function that smoothly transitions between the multi-class unhinged loss and the Mean Absolute Error (MAE). Our numerical experiments demonstrate its strong performance (section 8).

iii) We show the following results about the multi-class unhinged loss function:

a) It is the unique convex, non-trivial, non-increasing, multi-class symmetric loss function under the assumption of invariance to permutations (defined in section 3.1).

b) It is the linear approximation to any symmetric loss function at points with equal components (under proper assumptions, section 7).

c) It is the linear approximation to the cross-entropy loss function at points with equal components (section 7).

d) It is equal to the MultiMargin loss function (multi-class extension to the Hinge loss) in a region close enough to a point with equal components (section 6.3).

## 2 RELATED WORK

Natarajan et al. (2013) proposed a loss correction approach in the binary case which was extended to the multi-class case in Patrini et al. (2016b) by estimating a noise transition matrix. In order to facilitate the estimation of this transition matrix, Yao et al. (2020) considered a factorization of the matrix in two easier to estimate matrices. For its part, Li et al. (2021) estimated the transition matrix and learned the classifier simultaneously (end-to-end). In the terminology of Algan & Ulusoy (2021), the different approaches above belong to the family of noise model based methods since they try to estimate the noise structure directly. On the other hand, noise model free methods mainly try to design intrinsically robust loss functions or to exploit different forms of regularization. Our work is concerned with the problem of designing new robust loss functions (through a process of symmetrization of a loss function) and so it belongs to the family of noise model free methods. An advantage of such methods is that they can be computationally less costly since they do not require to estimate the noise model.

Ghosh et al. (2015) introduced the symmetry condition in the binary case and showed that it is a sufficient condition to make risk minimization robust to label noise. The sigmoid loss, the ramp loss and the probit loss satisfy this condition but the common convex loss functions do not. It turns out that the only binary convex loss function to be symmetric is the unhinged loss (van Rooyen et al., 2015). While the sigmoid loss, the ramp loss and the probit loss achieve robustness by reducing the impact of wrongly classified examples during training (more likely to be corrupted), the unhinged loss achieves robustness by increasing the impact of correctly classified examples during training (more likely to be clean) by being negatively unbounded.

Ghosh et al. (2017) proved label noise robustness results for multi-class symmetric loss functions. Under the symmetry condition, any minimizer of the risk on the corrupted distribution with uniform label noise is also a minimizer of the risk on the clean distribution. Noise tolerance under simple non uniform noise and class conditional noise are also obtained but under some more assumptions (the true risk for the optimal classifier must be 0). They propose the Mean absolute error (MAE) as a robust loss function for training neural networks. Their experimental results show that while the cross-entropy loss eventually severely fits the noise, the MAE is much more robust. However, the test performance on

clean data is better for the cross-entropy loss. The MAE loss is seen to be more prone to underfitting and to be slower to train because the gradient can saturate while training. Furthermore, the MAE loss does not necessarily outperforms the cross-entropy loss on noisy labels when early stopping is used with cross-entropy. The benefits of early stopping in the presence of noisy labels has been investigated in many works (Arpit et al., 2017), (Liu et al., 2020), (Song et al., 2020), (Li et al., 2020). Deep neural networks have a tendency to fit the training data with clean labels in the beginning of training and to memorize the corrupted labels later in training.

In order to exploit both the noise robustness of MAE and the speed of training of CE, Zhang & Sabuncu (2018) proposes a generalization of the two losses. Their loss function is the negative Box-Cox transformation and it allows to interpolate between the MAE and the CE with a hyperparameter. The symmetric cross-entropy loss is proposed in Wang et al. (2019). This loss function is composed of two terms. A robust term (reverse cross-entropy loss) and the standard cross-entropy loss (for convergence). The reverse cross-entropy loss is a scalar multiple of the MAE. Taylor cross entropy loss (Feng et al., 2021) considers approximations to the cross-entropy loss of different orders. The MAE is the first order Taylor approximation of CE. The second-order Taylor Series approximation of CE is an average combination of MAE and a lower bound of Mean Squared Error (MSE). Order 2 and above approximations of the CE are not symmetric loss functions. Considering different orders of approximation to the CE is another way to interpolate between the MAE and the CE. These different methods are therefore all trying to make a compromise between the robustness of the MAE and the better fitting ability of CE. Our method is always guaranteed to lead to a symmetric loss function (contrary to the approaches above) and the problem of underfitting can be alleviated by controlling the amount of saturation in the loss (with an hyperparameter in SGR and SGCE for example, section 6).

In the special case of the cross-entropy loss, our symmetrization method leads to a multi-class extension of the unhinged loss function which we call the multi-class unhinged. In most previous works, the unhinged loss in the multi-class setting was taken to be equivalent to the MAE (Zhang & Sabuncu, 2018), (Zhu et al., 2023). This is not the case in our work. Our multi-class unhinged loss is a linear symmetric loss function like its binary counterpart. This means that no softmax function is being used at the final layer. The work of Patrini et al. (2016b) considers a loss function that they call unhinged in their experiments in multi-class classification, but it is actually not a symmetric loss function. It is equivalent up to an additive constant and a multiplicative constant to minus the logit at the target label. Such a loss function is linear (no softmax function is being used) but not symmetric. The symmetrization of this loss function with our method leads also (like the symmetrization of the cross-entropy) to what we call the multi-class unhinged. Since these linear loss functions are unbounded, it is necessary to prevent numerical overflows. Patrini et al. (2016b) adds batch normalization to the final layer. Since this method is simple and leads to stable training, we also do the same.

The closest work to our own is Ma et al. (2020). They propose a normalization applicable to any loss function that leads to a symmetric loss function. They observe however that their robust loss functions often suffer from underfitting. The solution that they propose is called Active Passive Loss. An active loss function is only optimizing directly at the specified label. A passive loss also explicitly minimizes the probability for other classes. An Active passive loss is then defined to be a weighted combination of an active and a passive loss. Both the active and the passive loss are required to be robust in order to ensure that the combination is also robust.

The work of Patrini et al. (2016a) investigates binary loss functions that can be factorized as a sum of a class-insensitive term and a linear term. They refer to these losses as linear-odd losses (the odd part of the loss is linear). The labeled data centroid becomes the quantity of interest in the linear component of the loss and the subsequent works of Gao et al. (2016), Gong et al. (2021) and Gong et al. (2022) proposed estimators for this centroid when only corrupted data is available. Ding et al. (2022) extended these ideas to the multi-class case by decomposing the multi-class mean-squared error loss in order to also reduce the problem to centroid estimation. Our work considers instead the general decomposition of any multi-class loss function into a symmetric loss function and a class-insensitive term. Furthermore,

our main focus of investigation is the application of this decomposition to the cross-entropy loss and not the mean square error.

## 3 Preliminaries and assumptions

### 3.1 Assumptions on multi-class loss functions

We consider loss functions of the form $L(z, y)$, where $z = (z_i, \cdots, z_C) = h(x) \in \mathbb{R}^C$ is the last representation vector for some neural network, $x \in \mathbb{R}^d$ is an input and $y \in \{1, \cdots, C\}$ is a label for an example $(x, y)$ sampled from a distribution $D$ on $\mathbb{R}^d \times \{1, \cdots, C\}$. We will make two main assumptions on the loss function. First, we want the loss function $L(z, y)$ to be non-increasing.

**Definition 3.1.** *We say that $L(z, y)$ is a non-increasing multi-class loss function if for all $y$, the function $L(z, y)$ is a non-increasing function of $z_y$ when $z_k$ for $k \neq y$ is kept fixed.*

Secondly, we want a form of symmetry between the classes to hold (different from the notion of symmetry related to noise robustness). This is done to ensure that $L(z, k)$ and $L(z', k')$, where $k \neq k'$, will be the same function when the roles of $k$ and $k'$ are swapped. We will refer to this property as *invariance to permutations* and it is defined precisely below.

**Definition 3.2.** *The loss function $L(z, y)$ is invariant to permutations $\tau$ on $C$ elements if $L(\tau(z), \tau(y)) = L(z, y)$ for all $z$, $y$, and $\tau$. A permutation $\tau$ acts on $z$ by permuting the components of $z$, that is, $\tau(z) = (z_{\tau^{-1}(1)}, \cdots, z_{\tau^{-1}(C)})$.*

An example of a loss function satisfying both assumptions above is the standard cross-entropy loss defined by $L(z, y) = -\log(p_y)$, where $p_y = \frac{\exp(z_y)}{\sum_{k=1}^{C} \exp(z_k)}$ is obtained via the softmax function. Another simple example is the linear loss function $L(z, y) = -z_y$. An example of a loss function that does not satisfy the invariance to permutations property is $L(z, y) = -yz_y$.

### 3.2 Uniform label noise and the symmetry condition

Assume that with some probability $p$, instead of sampling the label of an example from the true distribution, we sample the label from a uniform distribution on the $C$ classes. In other words, define the corrupted distribution $\overline{D}$ to have the same marginal distribution as $D$ (that is $\overline{D}_x = D_x$) but with conditional distribution $\overline{D}_{y|x}$ given by

$$pU(\{1, \cdots, C\}) + (1 - p)D_{y|x},$$

where $U(\{1, \cdots, C\})$ is a uniform distribution over $\{1, \cdots, C\}$. Given a training $S$ from $D$, we can corrupt it to get a set $\overline{S}$ by changing an example $(x, y) \in S$ to $(x, \bar{y})$ $(\bar{y} \neq y)$ with probability $\frac{(C-1)p}{C}$.

It is then straightforward to get

$$L_{\overline{D}}(h) = \frac{p}{C}\left[\mathbb{E}_{x \sim D_x} \sum_{k=1}^{C} L(h(x), k)\right] + (1 - p)L_D(h),$$

where $L_D(h)$ is the expected loss (over distribution $D$) of classifier $h$. If we could get rid of the term $\mathbb{E}_{x \sim D_x} \sum_{k=1}^{C} L(h(x), k)$ above, the true risk on the clean distribution would be proportional to the true risk on the corrupted distribution (if $p < 1$). The optimizers of $L_{\overline{D}}(h)$ would then be the same as the optimizers of $L_D(h)$. This is the motivation for the symmetry condition.

**Definition 3.3.** *A loss function $L(z, y)$ is said to be symmetric if, for all $z$,*

$$\sum_{k=1}^{C} L(z, k) = constant.$$

## 4    THE GENERAL APPROACH

In order to define the symmetrization of a loss function, we ask the question of how to decompose a loss function into a sum of a symmetric loss function and a class-insensitive term. It happens to be the case that there is a unique such decomposition up to constants.

**Proposition 4.1.** *There is a unique (up to constants) decomposition of a loss function into a sum of a symmetric loss function and a class-insensitive term.*

Define $L^{sym}(z,y)$ by the following formula:

$$L^{sym}(z,y) = L(z,y) - \frac{1}{C}\sum_{k=1}^{C} L(z,k).$$

The loss function $L^{sym}(z,y)$ is symmetric and $L(z,y) - L^{sym}(z,y)$ is class-insensitive. The loss function $L^{sym}(z,y)$ is therefore the symmetric component inside the unique decomposition for $L(z,y)$ in Proposition 4.1.

## 5    CONDITIONAL DATA-DEPENDENT DIRICHLET PRIORS

In the case where the original loss function is the negative log-likelihood, our symmetrization method can be interpreted as a form of regularization induced from using Dirichlet priors on the outputs of the network. We explain precisely how below.

We consider neural networks having as outputs probability vectors on $C$ classes. That is, for a neural network with parameters $\theta$, the output of the neurak network on input $x$ is the conditional distribution $p(y|x,\theta)$. Let $\Delta_C := \{\pi = (\pi_1, \cdots, \pi_C) \mid \pi_i \geq 0 \text{ and } \sum_{i=1}^{C} \pi_i = 1\}$ be the probability simplex in dimension $C$. A neural network $f_\theta$ is then a function

$$f_\theta : \mathbb{R}^d \longrightarrow \Delta_C.$$

Suppose that we have a training set of $n$ i.i.d. pairs $(x_i, y_i)$ and denote by $X$ the $d \times n$ matrix obtained from aggregating the $n$ column vectors $x_i$. Also denote by $Y$ the column vector of training labels. In a Bayesian treatment, we would be interrested in the posterior distribution $p(\theta \mid X, Y)$. From Bayes rule, we get

$$p(\theta \mid X, Y) \propto p(Y \mid X, \theta)p(\theta \mid X).$$

It is commonly assumed that the prior is chosen completely independently of the training data, that is $p(\theta \mid X) = p(\theta)$. However, it is also possible to maintain the dependency on $X$, leading to the notion of a data-dependent prior. This data-dependent prior depends only on the observed covariates X and not on the observed response variables $Y$.

We want to define $p(\theta \mid X)$. Each $\theta$ represents a neural network $f_\theta$. Since we have access to $X$, we can look at $f_\theta(x) \in \Delta_C$ for $x$ in the training examples to define the prior. In Bayesian statistics, the Dirichlet distribution of order $C$ is often used as a distribution over $\Delta_C$ since it is the conjugate prior to the categorical distribution. It is defined by the density function

$$g(\pi\,; \alpha_1, \cdots, \alpha_C) = constant \times \prod_{i=1}^{C} \pi_i^{\alpha_i - 1},$$

where the parameters $\alpha_i$ satisfy $\alpha_i > 0$ for all $i$. If $\pi \in \Delta_C$ is distributed according to a Dirichlet distribution with parameters $\alpha = (\alpha_1, \cdots, \alpha_C)$, we will write $\pi \sim Dir(\alpha)$. A very natural first step to define our data-dependent prior $p(\theta \mid X)$ is to let $f_\theta(x_i) \sim Dir(\alpha(x_i))$ for each $x_i$ in the training set and where $\alpha(x)$ is a function of $x$. We then need to define a joint distribution over the vector $(f_\theta(x_1), \cdots, f_\theta(x_n))$. We will simply choose to have the $f_\theta(x_i)$'s mutually independent. We then have a joint distribution over the outputs of $f_\theta$ on the training data. This does not lead immediately to a distribution on $\theta$ however since many different $\theta$'s can have the same vector of outputs $(f_\theta(x_1), \cdots, f_\theta(x_n))$. We define an equivalence class on $\theta$ as follows:

$$[\theta]_X = [\theta']_X \text{ if and only if } f_\theta(x_i) = f_{\theta'}(x_i) \text{ for all } 1 \le i \le n.$$

So far, we have defined a distribution $p([\theta]_X \mid X)$. It is given by a product of Dirichlet distributions (to be technically correct, we have to take the restriction of this product of Dirichlet to the subset of $\Delta_C^n$ that can be realized with the hypothesis class $\{f_\theta\}$). We can then write

$$p(\theta \mid X) = \int_{[\theta']_X} p(\theta \mid [\theta']_X, X) p([\theta']_X \mid X) d[\theta']_X = p(\theta \mid [\theta]_X, X) p([\theta]_X \mid X).$$

We are therefore left with defining a distribution for $\theta$ inside of its equivalence class i.e. $p(\theta \mid [\theta]_X, X)$. A possible example would be to choose a uniform distribution. This would lead to $p(\theta \mid [\theta]_X, X) = \frac{1}{m([\theta]_X)}$, where $m([\theta]_X)$ is the measure of the set $\{\theta' \ s.t. \ [\theta']_X = [\theta]_X\}$. Up to an additive constant the negative log posterior is then given by

$$\sum_{i=1}^{n} -\log(p(y_i \mid x_i, \theta)) + \sum_{i=1}^{n} \sum_{k=1}^{C} -(\alpha_k(x_i) - 1)\log(p(k \mid x_i, \theta)) - \log(p(\theta \mid [\theta]_X, X)).$$

If we denote by $l(h(x), y)$ the negative log likelihood loss $(l(h(x), y) = -\log(p(y \mid h(x))))$, where $h(x)$ is the last representation vector) and if we drop the extra regularization term $-\log(p(\theta \mid [\theta]_X, X))$, then the quantity above is the sum over all examples of

$$l_{Dir}(h(x), y) := l(h(x), y) + \sum_{k=1}^{C} (\alpha_k(x) - 1)l(h(x), k).$$

**Lemma 5.1.** *Assume that $\alpha_k(x) = \alpha$ is constant. Then, the loss $l_{Dir}$ is symmetric if and only if $\alpha = \frac{C-1}{C}$ or $l$ is already symmetric.*

The special case of $l_{Dir}$ with $\alpha_k(x) = \frac{C-1}{C}$ leads to the same loss as $l^{sym}$, that is, the symmetrization of $l$. An illustration for the binary case (Beta distributions) is given in Figure 1 (Appendix D).

## 6 Symmetrization of different loss functions

### 6.1 Symmetrization of the cross-entropy loss

It is easy to verify that the symmetrization of the multiclass cross-entropy loss $(L(z, y) = -\log(p_y)$, where $p_y = \frac{\exp(z_y)}{\sum_{k=1}^{C} \exp(z_k)}$ is obtained via the softmax function) is the linear function

$$-z_y + \frac{1}{C} \sum_{k=1}^{C} z_k.$$

We call this loss function the multi-class unhinged loss. We note that since $-\log(p_y) = -z_y + \log(\sum_{k=1}^{C} \exp(z_k))$ and the term $\log(\sum_{k=1}^{C} \exp(z_k))$ is class-insensitive, the symmetrization of the cross-entropy loss is the same as the symmetrization of the linear loss $-z_y$. The multi-class unhinged loss satisfies the same fundamental result as the binary unhinged loss.

**Theorem 6.1.** *The multi-class unhinged loss is the unique convex, non-trivial, non-increasing, multi-class symmetric loss function satisfying the property of invariance to permutations (up to an additive and a multiplicative constant).*

**Remark 6.2.** *Without the property of invariance to permutations, the uniqueness result would not be true. Indeed, consider the following example with 3 classes:*

$$L(z, y) = \begin{cases} -z_1 + z_2 + z_3 & \text{if } y=1 \\ -z_2 + z_1 & \text{if } y=2 \\ -z_3 & \text{if } y=3 \end{cases}$$

*This loss function is convex (actually linear) for all $y$, non-increasing and symmetric. However, it is not equivalent to the multi-class uninged loss function.*

## 6.2 Symmetrization of the mean square error

The mean square error for classification is given by

$$L(z, y) = ||e_y - s(z)||_2^2 = ||s(z)||_2^2 + 1 - 2p(y|z),$$

where $s$ is the softmax function and the $y^{th}$ coordinate of $s(z)$ is $p(y|z)$. Since the term $||s(z)||_2^2$ is independent of the label, the symmetrization operator removes it and we are left with a loss equivalent to the MAE.

## 6.3 symmetrization of the Generalized ramp loss functions (SGR)

The MultiMargin loss function is a multi-class extension of the Hinge loss. It is defined by

$$l_\gamma(z, y) = \sum_{k \neq y} \max\{0, \gamma - (z_y - z_k)\},$$

where $\gamma > 0$ is the margin parameter. We propose further to investigate a multi-class extension to the Ramp loss defined by

$$R_\gamma(z, y) := l_\gamma(z, y) - l_{-\gamma}(z, y).$$

It is possible to generalize (and interpolate) between the MultiMargin and the Ramp loss by adding an hyperparameter $\beta$:

$$R_\gamma^\beta(z, y) := l_\gamma(z, y) - \beta l_{-\gamma}(z, y).$$

We will call $R_\gamma^\beta(z, y)$ the generalized Ramp loss and the symmetrization of the generalized Ramp loss will be abreviated by SGR.

**Proposition 6.3.** *Assume that $z$ satisfies $|z_k - z_k'| \leq \gamma$ for all $1 \leq k, k' \leq C$. Then, $R_\gamma^\beta(z, y)$ and SGR are equivalent to the multi-class unhinged loss function.*

When the margin is large enough or the representation vector is constrained close enough to a vector with equal components, the generalized ramp loss (and so also the MultiMargin loss function) is robust since it is equivalent to the multi-class unhinged loss function. Figure 2 in Appendix D illustrates these loss functions in the binary case.

## 6.4 Symmetrization of the generalized cross-entropy loss (SGCE)

The generalized cross-entropy loss (GCE) (Zhang & Sabuncu, 2018) is defined by

$$L_q(z, y) := \frac{1 - p(y|z)^q}{q},$$

where $q \in (0, 1]$. When $q$ goes to 0, the loss converges to the cross-entropy. When $q = 1$, we get the MAE. The symmetrization of the generalized cross-entropy loss (SGCE) leads therefore to a form of interpolation between the multi-class unhinged and the MAE. The unhinged loss function is robust by maintaining larger gradients for examples already correctly classified. The MAE is robust by reducing the gradient of incorrectly classified examples (since they might be corrupted). The SGCE loss function allows to realize a trade-off between these two strategies. Figure 3 in Appendix D illustrates GCE and SGCE in the binary case.

## 7 Linear approximations of multi-class loss functions

Since not any linear loss function is symmetric in the multi-class setting, it is an interesting question to ask when the linear approximation around a point $z'$ (for example the origin) is

symmetric, that is equivalent to the multi-class unhinged loss function. If this is the case, we can expect the loss function to be robust when training stays around $z'$. Regularization methods like weight decay, batch normalization and early stopping can then all contribute to maintain training in the robust region.

It happens to be the case that the linear approximation of the cross-entropy loss around $z = z'$ where the components of $z'$ are all equal is equivalent to the multi-class unhinged loss function. Indeed, the gradient of the cross-entropy loss with respect to $z$ is given by

$$\nabla_z L(z, y) = s(z) - y,$$

where $s(z)$ is the output of the softmax function evaluated at $z$ and $y$ is the one-hot encoding for the class. The linear approximation of $L(z, y)$ at $z'$ is then given by

$$(s(z') - y)^{tr}(z - z') + L(z', y).$$

Up to constants from the point of view of the variable $z$, we only need to consider $(s(z') - y)^{tr}z$. Since $s(z') = (\frac{1}{C}, \cdots, \frac{1}{C})$ when all the components of $z'$ are equal,

$$(s(z') - y)^{tr}z = (\frac{1}{C} - 1)z_y + \frac{1}{C}\sum_{k \neq y} z_k = -z_y + \frac{1}{C}\sum_{k=1}^{C} z_k.$$

We conclude that the linear approximation of the cross-entropy loss around $z'$ is equivalent to the multi-class unhinged loss function. The cross-entropy loss is therefore "locally symmetric" around any such $z'$. This can help to explain why the cross-entropy loss can already be somewhat robust in particular when early stopping is being used. Indeed, if training stays close enough to an initial point such that the probability for each class is the same, we are approximately training with a symmetric loss function.

We found an example of a non-robust loss function that happens to be locally equivalent to the multi-class unhinged loss function at some specific points. If the loss function is globally robust, it is actually guaranteed to be locally equivalent to the multi-class unhinged loss function at every point $z'$ with equal components (that is not a critical point).

**Proposition 7.1.** *Assume that $L(z, y)$ is non-increasing, symmetric, satisfies the property of invariance to permutations and is differentiable at $z'$ a vector with equal components. Then, the linear approximation of $L(z, y)$ at $z'$ is equivalent to the multi-class unhinged loss function if $\nabla_z L(z, y)|_{z=z'} \neq 0$.*

The previous result helps in understanding the relevance of the multi-class unhinged loss function among differentiable multi-class symmetric loss functions. The linear approximations of the these loss functions at points with equal components cannot be any linear function (like in the binary case), it must have specific coefficients as prescribed by the multi-class unhinged loss function.

## 8 EXPERIMENTS

### 8.1 METHOD

We compare the performance of different robust loss functions on CIFAR10 (Krizhevsky, 2009). We train a resnet44 (He et al., 2016) for 80 epochs with stochastic gradient with momentum. The momentum is fixed to 0.9 and weight decay is fixed to 0.001. The learning rate is chosen inside $\{0.01, 0.1, 1.0\}$ and is decayed by a factor 10 at 60% of training and again by another factor 10 at 90% of training. We tune the learning rates and the different hyperparameters of the different loss functions on a validation set consisting in 5000 examples taken from the original training set. This validation set is also used for early stopping (early stopping is done on every method).

Since the multi-class unhinged loss function is negatively unbounded (and also other loss functions resulting from the symmetrization operation can be negtively unbounded), we

Table 1: Accuracy (mean of 3 runs with standard deviation in parentheses) of robust loss functions on CIFAR10 with probabilities of corruption $p \in \{0.2, 0.4, 0.6, 0.8, 0.95\}$. The 4 best results are in bold.

|  | 0.2 | 0.4 | 0.6 | 0.8 | 0.95 |
|---|---|---|---|---|---|
| MAE | 84.23 (0.15) | 83.01 (0.23) | 78.87 (0.31) | 60.83 (0.22) | **26.29 (0.27)** |
| CE | 88.33 (0.19) | 85.05 (0.26) | 79.33 (0.35) | 67.80 (0.61) | 22.80 (0.99) |
| NCE | 88.72 (0.12) | 84.34 (0.19) | 79.83 (0.16) | 67.09 (1.08) | 22.69 (0.49) |
| Unhinged | 88.74 (0.18) | 85.98 (0.07) | 80.61 (0.23) | 69.06 (0.42) | **26.42 (0.58)** |
| SGR | **88.99 (0.23)** | **86.18 (0.12)** | 81.22 (0.18) | **71.28 (0.39)** | 24.28 (0.60) |
| MAE+NCE | 87.98 (0.04) | 85.71 (0.11) | **82.12 (0.14)** | **71.49 (1.01)** | **26.77 (0.49)** |
| GCE | **89.23 (0.54)** | **86.57 (0.18)** | **82.31 (0.25)** | 70.23 (0.89) | 24.96 (1.05) |
| NGCE | **89.29 (0.04)** | **86.34 (0.20)** | **82.16 (0.31)** | **71.47 (0.65)** | 25.13 (0.55) |
| SGCE | **89.22 (0.07)** | **86.29 (0.31)** | **82.31 (0.20)** | **72.25 (0.52)** | **27.66 (0.30)** |

need a method to prevent numerical overflows and promote stable training. Adding a batch normalization layer to the final layer allows for successful training of our symmetrized loss functions (every other layer is already using batch normalization). Since there can be benefits in adding this last batch normalization layer for other methods also (see Table 2 in Appendix C), we add this batch normalization layer for every method in order to ensure fairness of comparison.

For the SGR loss, we fix the margin to 10 and tune $\beta$ in $\{0, 0.5, 1, 3\}$. For GCE, the normalized GCE (NGCE) and the symmetrized GCE (SGCE) losses, we tune $q$ in $\{0.25, 0.50, 0.75\}$ ($q = 0$ and $q = 1$ are already represented in Table 1). For MAE+NCE (Ma et al., 2020), we consider an hyperparameter $\alpha$ and the loss $(1 - \alpha)\ NCE + \alpha\ MAE$. The hyperparameter $\alpha$ is chosen in $\{0.25, 0.50, 0.75\}$ ($\alpha = 0$ and $\alpha = 1$ are already represented in Table 1). We report the mean and the standard deviation over 3 runs for probabilities of corruption (see section 3.2) $p \in \{0.2, 0.4, 0.6, 0.8, 0.95\}$ in Table 1. The clean data baseline is given in Table 3 in Appendix C.

## 8.2 Discussion of results

We first note that the cross-entropy loss can already be reasonably robust in a context with low amount of noise. This makes sense in the light that with enough regularization (e.g. batch normalization, early stopping), the cross-entropy can be approximately symmetric (approximately equivalent to the multi-class unhinged loss). However, the gap with the best performing methods increases as the probability of corruption increases. The symmetrized cross-entropy (multi-class unhinged) performed better than the cross-entropy and the normalized cross-entropy in the presence of noise. However, it has a tendency to under performed compared to more sophisticated methods that can control the amount of saturation. Our symmetrization of the generalized cross-entropy loss performed very well on every noise rate.

## 9 Conclusion

This work proposed a principled symmetrization method for designing robust loss functions to label noise. The method is related to regularization from conditional data-dependent Dirichlet priors. The symmetrization of the categorical cross-entropy loss leads to the unique convex, non-trivial, non-increasing multi-class symmetric loss function under the technical assumption of invariance to permutations. As such, this loss function extends the binary unhinged loss and so, we refer to it as the multi-class unhinged loss function. Empirical results showed that our symmetrized loss functions can be trained by adding a batch normalization layer to the final layer. The symmetrization of the generalized cross-entropy leads to particularly good results overall. We hope that our approach will further help in the understanding and the conception of new loss functions robust to label noise in the future.

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

## A   Decomposition in the binary case and Taylor series

In the binary case, the unique decomposition of a loss function $\phi(z)$ into a sum of a symmetric loss function and a class-insensitive loss function is the sum of its odd part $\frac{\phi(z)-\phi(-z)}{2}$ with its even part $\frac{\phi(z)+\phi(-z)}{2}$. If $\phi$ admits a Taylor series expansion around 0, it is possible to characterize the symmetry condition using the coefficients of the Taylor series and to express the odd part and the even part with the series.

**Proposition A.1.** *Assume that $\phi$ is an infinitely differentiable potential and without loss of generality that $\phi(0) = 0$. Then, $\phi$ is symmetric if and only if $\phi^{(k)}(0) = 0$ for all $k$ even. That is, $\phi$ is symmetric if and only if the even coefficients of its Taylor expansion at $0$ are all $0$.*

Since the odd part of $\phi$ is the sum over the terms with odd coefficients of the Taylor series, our symmetrization method corresponds to the very natural process of keeping the odd coefficients of the original loss and replacing the even coefficients by 0's. Every truncation of the Taylor expansion for the symmetric loss function is also symmetric. This allows approximating any such symmetric loss functions with simpler polynomial symmetric loss functions.

The decomposition as a sum of an odd and an even function in the binary case makes sense because changing the label amounts to changing the sign of $z$. However, this does not generalize immediately to the multi-class case. When taking the point of view that the odd binary loss functions are the symmetric loss functions and that the even binary loss functions are the class-insensitve loss functions, we can get a decomposition that holds in the multi-class case also.

## B   Proofs

**Proof of Proposition 4.1:** Suppose $l = l_1^{sym} + l_1^{ins} = l_2^{sym} + l_2^{ins}$. That is, we have two decompositions of $l$ as a sum of a symmetric and a class-insensitive loss function. Then,

$$l_1^{sym} - l_2^{sym} = l_2^{ins} - l_1^{ins}.$$

Since $l_1^{sym} - l_2^{sym}$ is symmetric and $l_2^{ins} - l_1^{ins}$ is class-insensitive, they must be both symmetric and class insensitive. The only loss functions that are both class insensitive and symmetric are constant. Indeed, if an arbitrary loss function $l'(z, y)$ is both symmetric and class-insensitive, then

$$constant = \sum_{y=1}^{C} l'(z, y) = Cl'(z, k)$$

for any $1 \leq k \leq C$ and any $z \in \mathbb{R}^C$, and therefore $l'(z, y)$ is constant. We conclude that $l_1^{sym}$ is equal to $l_2^{sym}$ up to an additive constant and also $l_1^{ins}$ is equal to $l_2^{ins}$ up to an additive constant.■

In order to prove the uniqueness result among convex functions for the multi-class unhinged loss, we start by proving uniqueness among linear functions. The invariance to permutations property is crucial and the next Lemma proves some constraints that must hold on the coefficients of a linear loss satisfying the invariance to permutations property.

**Lemma B.1.** *Assume that a linear loss $L(z,y) = \sum_{k=1}^{C} a_k(y)z_k$ satisfies the invariance to permutations property. Then,*

    *i)* $a_y(y) = a_k(k)$ *for all* $k, y$.

    *ii)* $a_k(y) = a_y(k)$ *for all* $k, y$.

    *iii)* $a_k(y) = a_{k'}(y)$ *if* $k \neq y$ *and* $k' \neq y$.

**Proof:**

    i) For given $k$ and $y$, consider a permutation $\tau$ switching $k$ and $y$. From the invariance to permutations property, $L(z,y) = L(\tau(z), \tau(y)) = L(\tau(z), k)$ for all $z$. Pick $z = e_y$. Then,

$$a_y(y) = L(e_y, y) = L(\tau(e_y), k) = L(e_k, k) = a_k(k).$$

    ii) Consider $\tau$ as above, but now pick $z = e_k$. Then,

$$a_k(y) = L(e_k, y) = L(\tau(e_y), \tau(k)) = L(e_y, k) = a_y(k).$$

    iii) Fix $y$ and let $k \neq y$ and $k' \neq y$. Consider any permutation $\tau$ with fixed point $y$ satisfying $\tau(k) = k'$ and $\tau(k') = k$. From the invariance to permutations property, $L(z,y) = L(\tau(z), \tau(y)) = L(\tau(z), y)$. Since any two linear functions equal everywhere must have the same coefficients, from the equality $L(\tau(z), y) = L(z, y)$ and noting that the coefficient of $z_k$ in $L(z,y)$ is $a_k(y)$ and the coefficient of $z_k$ in $L(\tau(z), y)$ is $a_{k'}(y)$, we conclude that $a_k(y) = a_{k'}(y)$.

**Proposition B.2.** *The multi-class unhinged loss is the unique (up to a multiplicative constant) non-trivial, non-increasing, linear multi-class symmetric loss function that satisfies the property of invariance to permutations.*

**Proof:** It is easy to verify that the multi-class unhinged loss function is a symmetric, non-increasing, linear loss function that satisfies the invariance to permutations property. We now show that it is the unique such loss function up to a multiplicative constant. Let $L(z,y) = \sum_{k=1}^{C} a_k(y)z_k$. Using the symmetry condition and rearranging terms leads to

$$\sum_{k=1}^{C} \left( \sum_{y=1}^{C} a_k(y) \right) z_k = constant,$$

for all $z$. If a linear function is constant, all its coefficients must be 0 and so

$$\sum_{y=1}^{C} a_k(y) = 0,$$

for all $k$. Using Lemma B.1 *ii)* leads to $\sum_{y=1}^{C} a_y(k) = 0$, for all $k$. For convenience, we change the names of the indices in the previous equality and write

$$\sum_{k=1}^{C} a_k(y) = 0,$$

for all $y$. From the assumption that $L(z,y)$ is non-increasing, we must have $a_y(y) \leq 0$. If $a_y(y) = 0$, then $\sum_{k \neq y} a_k(y) = 0$. But, from Lemma B.1 *iii)*, we must then have $(C-1)a_k(y) = 0$ for any $k$. The loss $L(z,y)$ would then be identically 0. Therefore, for the loss to be non-trivial, we must have $a_y(y) < 0$. Since we consider the loss up to a multiplicative constant, we can assume that $a_y(y) = -1$ for all $y$ (this holds for all $y$ from Lemma B.1 *i)*). Finally, from Lemma B.1 *iii)*, $a_k(y) = \frac{1}{C-1}$ if $k \neq y$. Consequently,

$$L(z,y) = -z_y + \frac{1}{C-1} \sum_{k \neq y} z_k = \frac{C}{C-1} \left( -z_y + \frac{1}{C} \sum_{k=1}^{C} z_k \right).$$

This concludes the proof showing that $L(z,y)$ is equal to the multi-class unhinged loss function up to a multiplicative constant.■

**Proof of Theorem 6.1:** Assume that $L(z,y)$ is a convex function of $z$. From the symmetry condition, we get

$$L(z,y) = constant - \sum_{k \neq y} L(z,k).$$

Since a sum of convex functions is convex, $-\sum_{k \neq y} L(z,k)$ is concave. It follows that $L(z,y)$ is both convex and concave. The only functions that are both convex and concave are affine functions. Therefore, under the assumptions of the theorem, it follows from Proposition B.2 that $L(y,z)$ must be equal to the multi-class unhinged loss function up to an additive and a multiplicative constant. ■

**Proof of Lemma 5.1:** If $\alpha_k(x) = \alpha$, we have

$$\sum_{y=1}^{C} l_{Dir}(h(x),y) = \left[ 1 + C(\alpha - 1) \right] \sum_{y=1}^{C} l(h(x),y).$$

Therefore, if $l$ is not already symmetric, we must have $1 + C(\alpha - 1) = 0$ for $l_{Dir}$ to be symmetric. That is, we must have $\alpha = \frac{C-1}{C}$.

**Proof of Proposition 6.3:** Under the assumption that $z$ satisfies $|z_k - z'_k| \leq \gamma$ for all $1 \leq k, k' \leq C$, we have that $\max\{0, \gamma - (z'_k - z_k)\} - \beta \max\{0, -\gamma - (z'_k - z_k))\}$ is equal to $\gamma - (z'_k - z_k)$. Therefore,

$$\begin{aligned} R_\gamma^\beta(z,y) &= \sum_{k \neq y} \left( \gamma - (z_y - z_k) \right) \\ &= (C-1)\gamma - (C-1)z_y + \sum_{k \neq y} z_k \\ &= (C-1)\gamma - Cz_y + \sum_{k=1}^{C} z_k. \end{aligned}$$

Up to an additive and a multiplicative constant, this last quantity is equal to the multi-class unhinged loss. The symmetrization of $R_\gamma^\beta(z,y)$ will also satisfy the same property since the symmetrization of an already symmetric loss function falls back on itself and $R_\gamma^\beta(z,y)$ is symmetric in the region given by our assumption as shown above.

**Proof of Proposition 7.1:** We first need to show that $\left[ \nabla_z L(z,y)|_{z=z'} \right]^{tr} z$ satisfies the property of invariance to permutations. Let $\tau$ be a permutation and $T$ the corresponding permutation matrix. From the chain rule and the invariance to permutations property for $L(z,y)$, we get

$$\begin{aligned} \left[ \nabla_z L(z,\tau(y))|_{z=z'} \right]^{tr} \tau(z) &= \left[ \nabla_z L(\tau^{-1}(z),y)|_{z=z'} \right]^{tr} \tau(z) \\ &= \left[ \nabla_z L(z,y)|_{z=\tau^{-1}(z')} \right]^{tr} T^{-1} \tau(z) \\ &= \left[ \nabla_z L(z,y)|_{z=z'} \right]^{tr} z, \end{aligned}$$

where the last line is true since $\tau^{-1}(z') = z'$ when all the components of $z'$ are equal. We conclude that $\left[ \nabla_z L(z,y)|_{z=z'} \right]^{tr} z$ satisfies the property of invariance to permutations.

We now need to show that $\left[ \nabla_z L(z,y)|_{z=z'} \right]^{tr} z$ satisfies the symmetry condition. Differentiating both sides of the symmetry condition for $L(z,y)$ with respect to $z$ at $z'$ leads to

$$\sum_{k=1}^{C} \nabla_z L(z,k)|_{z=z'} = 0.$$

Taking the dot product with $z$ on both sides leads to the conclusion that $\left[\nabla_z L(z,y)|_{z=z'}\right]^{tr} z$ is symmetric. From the uniqueness result for the multi-class unhinged loss, the proposition follows if $\left[\nabla_z L(z,y)|_{z=z'}\right]^{tr} z$ is non-trivial, that is, if $\nabla_z L(z,y)|_{z=z'} \neq 0$. ∎

**Example B.3.** *When a differentiable loss function is globally symmetric, it is also locally symmetric everywhere. However, it need not be equivalent to the multi-class unhinged loss everywhere. Indeed, even if the loss function satisfies the property of invariance to permutations, the linear approximation might not satisfy the same property everywhere. An example is the MAE in three variables:*

$$L(z,y) = 2 - 2 \frac{\exp(z_y)}{\sum_{k=1}^{3} \exp(z_k)}.$$

*If we let $l(z,y) = \left[\nabla_z L(z,y)|_{z=z'}\right]^{tr} z$ with $z' = (1,0,0)$, we get*

$$l(z,y) = \frac{2}{(e+2)^2} \begin{cases} (-2e, e, e) \cdot z & \text{if y=1} \\ (e, -e-1, 1) \cdot z & \text{if y=2}. \\ (e, 1, -e-1) \cdot z & \text{if y=3} \end{cases}$$

*This loss is symmetric as it should be. However, it does not satisfy the property of invariance to permutations and it is not equivalent to the multi-class unhinged loss function.*

## C    CROSS-ENTROPY BASELINE AND CLEAN DATA BASELINE

Some improvements can be made to the cross-entropy baseline by adding a batch normalization layer to the final layer of the network.

Table 2: Accuracy (mean of 3 runs with standard deviation in parentheses) of the cross-entropy baseline with and without batch normalization at the final layer with probabilities of corruption $p \in \{0.0, 0.2, 0.4, 0.6, 0.8, 0.95\}$ on CIFAR10.

|        | 0.0          | 0.2          | 0.4          | 0.6          | 0.8          | 0.95         |
|--------|--------------|--------------|--------------|--------------|--------------|--------------|
| CE     | 91.22(0.55)  | 87.88(0.18)  | 84.37(0.65)  | 79.08(0.64)  | 56.76(1.68)  | 22.61(0.82)  |
| CE+BN  | 92.37(0.17)  | 88.33(0.19)  | 85.05(0.26)  | 79.33(0.35)  | 67.80(0.61)  | 22.80(0.99)  |

Table 3: Accuracy (mean of 3 runs with standard deviation in parentheses) of robust loss functions on CIFAR10 with clean data. The 4 best results are in bold.

| MAE      | 85.84 (0.23)     |
|----------|------------------|
| CE       | **92.37 (0.17)** |
| NCE      | **91.87 (0.21)** |
| Unhinged | 91.49 (0.08)     |
| SGR      | 91.36 (0.12)     |
| MAE+NCE  | 89.36 (0.20)     |
| GCE      | **92.07 (0.08)** |
| NGCE     | **91.66 (0.23**) |
| SGCE     | 91.16 (0.24)     |

## D    ILLUSTRATIONS IN THE BINARY CASE

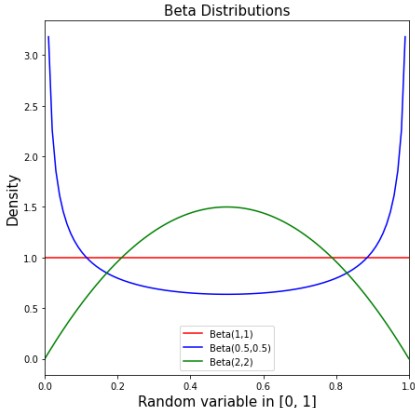

Figure 1: The added regularization (induced from a $Beta(\frac{1}{2}, \frac{1}{2})$ prior in the binary case) favours probability vectors with less entropy.

Figure 2: The generalized Ramp loss in (a) and its symmetrization in (b).

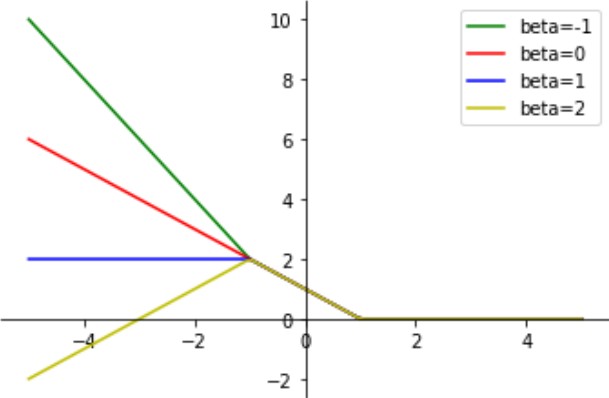

(a) For $\beta = 0$ in the generalized Ramp loss, we recover the Hinge loss and for $\beta = 1$ the Ramp loss.

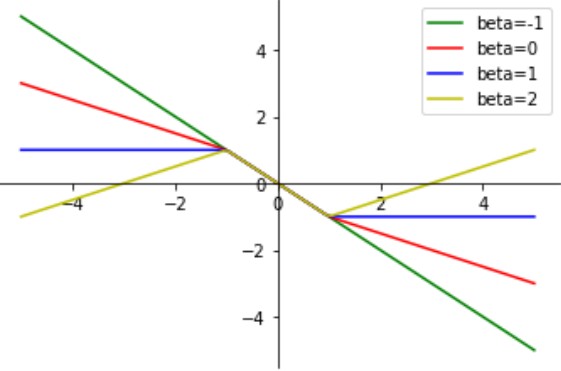

(b) Varying $\beta$ allows to obtain different levels of saturation in the symmetrization of the generalized Ramp loss.

Figure 3: The generalized cross-entropy loss in (a) and its symmetrization in (b).

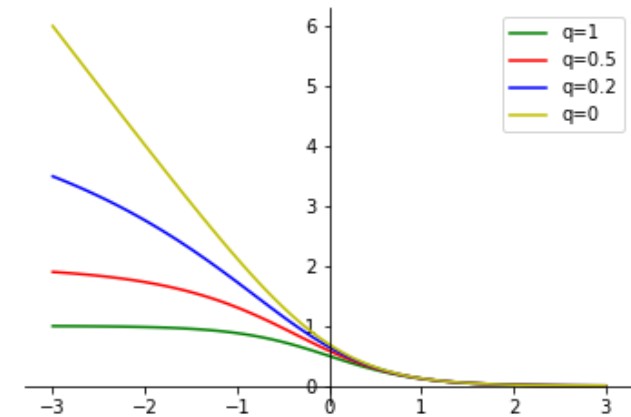

(a) GCE is not symmetric but becomes more robust as $q$ approches 1.

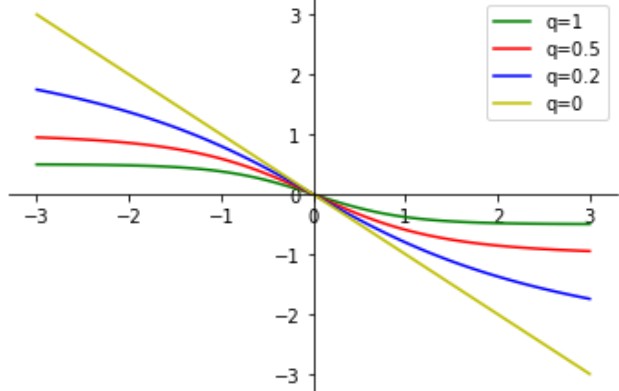

(b) SGCE is always symmetric and $q$ controls the amount of saturation in the loss by interpolating between the unhinged loss and the MAE.

