# OpenReview forum: "Symmetrization of Loss Functions for Robust Training of Neural Networks in the Presence of Noisy Labels and the Multi-class Unhinged Loss Function"
_ICLR.cc/2024/Conference — ICLR 2024 Conference Withdrawn Submission_

### Official Review · Reviewer_81Mq · 2023-10-30

**Soundness:** 1 poor
**Presentation:** 2 fair
**Contribution:** 1 poor
**Rating:** 3
**Confidence:** 5

**Summary:**

This work introduces a very simple method for symmetrizing a loss function and explains it as a Dirichlet prior. It further demonstrates that the unhinged loss is the only convex symmetric loss function for multi-class classification. Additionally, it symmetrizes specific loss functions, such as SGR and SGCE, and conducts some preliminary experiments on CIFAR-10 with symmetric label noise.

**Strengths:**

- The symmetrization of loss functions is very straightforward.
- The paper presents its concepts in a clear manner and is easy to follow.

**Weaknesses:**

- The paper template or compilation environment may have issues, leading to the use of fonts that are not in line with the ICLR reference fonts, potentially not meeting submission requirements
- The proposed symmetrization is equivalent to negative label smoothing, which resembles the work presented in [1]. The authors should provide a more in-depth discussion of the novelty of their approach, including its relationship to other existing works [2,3,4].
- The unhinged loss mentioned by the authors is not novel enough [5, 6].
- Although the method proposed by the authors is straightforward, the introduction of the negative part in the symmetrization of losses could potentially result in an unstable training process. I conducted some experiments based on code from [7], suggesting significant issues, including the cccurance of "nan" with CE and non-convergence of other losses. The authors should conduct additional experiments and provide further analysis on this matter.
- The experimental section of the paper is rather limited, focusing solely on basic experiments conducted on CIFAR-10. This level of experimentation is insufficient to support the paper's acceptance.
- The writting style is not very fluent, and a reorganization of the structure is recommended. For example, it is strongly advised to move the content from Section 6 to Section 4.



[1] Wei J, Liu H, Liu T, et al. To Smooth or Not? When Label Smoothing Meets Noisy Labels[C]//International Conference on Machine Learning. PMLR, 2022: 23589-23614.

[2] Zhou X, Liu X, Jiang J, et al. Asymmetric loss functions for learning with noisy labels[C]//International conference on machine learning. PMLR, 2021: 12846-12856.

[3] Liu Y, Guo H. Peer loss functions: Learning from noisy labels without knowing noise rates[C]//International conference on machine learning. PMLR, 2020: 6226-6236.

[4] Charoenphakdee N, Lee J, Sugiyama M. On symmetric losses for learning from corrupted labels[C]//International Conference on Machine Learning. PMLR, 2019: 961-970.

[5] Van Rooyen B, Menon A, Williamson R C. Learning with symmetric label noise: The importance of being unhinged[J]. Advances in neural information processing systems, 2015, 28.

[6] Long P M, Servedio R A. The perils of being unhinged: On the accuracy of classifiers minimizing a noise-robust convex loss[J]. Neural Computation, 2022, 34(6): 1488-1499.

[7] Ma X, Huang H, Wang Y, et al. Normalized loss functions for deep learning with noisy labels[C]//International conference on machine learning. PMLR, 2020: 6543-6553.

**Questions:**

Please see weakness.

---

> ### Author Response · Authors · 2023-11-22
> **Link with negative label smoothing**
>
> Thank you for your comments.
>
> Link with negative label smoothing: If we denote by $L^r(z,y)$ the loss obtained from smoothing the cross-entropy loss with rate $r$, the multi-class unhinged is the limiting loss function when smoothing goes to negative infinity. We have
>
> $lim_{r \rightarrow -\infty} \frac{L^r(z,y)}{1-r}=$multi-class unhinged.
>
> Another way to see this (following from the relation between label smoothing and noise correction) is that the multi-class unhinged is obtained as the limit when the probability of corruption $p$ goes to $1$ in the noise correction loss. When using a symmetric loss function, we do not need to know $p$ since robustness is obtained for any $p$. The multi-class unhinged is like choosing $p=1$ and is robust for any $p$.
>
> Link with peer loss function: When the prior probability of y is uniform, the expected peer loss (with CE) is equal to the expected multi-class unhinged over the corrupted distribution. For each sample, the peer loss requires to draw two other samples (the peer samples). These additional samples are not needed when using the symmetrization of a loss function however.
>
> -Numerical stability: As we explained, we could solve the problem of 'nan' (the loss is negatively unbounded) by adding a batch normalization layer to the final layer of the network. Another idea to increase even further numerical stability is to remove the learnable affine parameters in this last batch normalization layer (for example by setting 'affine=False' in pytorch).

---

### Official Review · Reviewer_FQCN · 2023-11-06

**Soundness:** 3 good
**Presentation:** 2 fair
**Contribution:** 2 fair
**Rating:** 5
**Confidence:** 4

**Summary:**

The manuscript considers loss functions for multi-class classification in the presence of uniform label noise. It is known that a certain notion of symmetry in the loss functions (commonly used losses satisfy the property) is central to being robust against uniform label noise --- i.e., the bayes optimal classifiers for the underlying clean distribution and the noisy distribution (where some labels are flipped) coincide. In this work, the authors make a simple observation that any loss function can be decomposed as a sum of a symmetric component and a label-insensitive component, that is unique up to constant factor. They apply this observation to the standard cross-entropy loss and derive a multi-class version of the so-called unhinged loss studied in Rooyen et al 2015, which is symmetric and therefore robust to (uniform) label noise. The authors show interesting properties of the loss function, and its connection to robustness of SGD-based optimization.

Overall, the paper develops and presents a few key ideas of merit, but I felt a) the contributions are a bit short of a strong ML venue, b) the paper is a bit all over the place, and the core contributions/messages don't stand out clearly, c) the paper's organization and writing can be improved a lot, and the current version hampers clarity.

**Strengths:**

- Rigorous development of results, several interesting properties proved and connections made.
- A simple but powerful observation in Proposition 4.1, that leads to symmetrization results for CE loss and generalized CE loss, and derivations of new robust loss functions for multi-class classifications.
- Connections to SGD-based optimization + early stopping of standard CE loss

**Weaknesses:**

- Applicability to (only) uniform label noise is somewhat limiting. What are the ideas in the paper that could be extended/adapted to more general noise models?
- Lack of clarity in presentation/writing
- Take-away messages are not clear. If unhinged multi-class loss function is indeed what practitioners should consider in some if not all scenarios, we need to see a lot more empirical support than the results in Section 8.

**Questions:**

- Please respond to the points raised under 'weaknesses'
- Section 5 is confusing. Are there $l_{Dir}(.,.)$ forms that are meaningful _and_ symmetric besides the one in Lemma 5.1?

---

> ### Author Response · Authors · 2023-11-22
> **More general noise models**
>
> Thank you for your comments.
>
> -More general noise models: it is possible to extend to symmetry condition to a generalized symmetry condition by choosing a different distribution in the definition of symmetry. We want to point out first that symmetric loss functions can also exhibit robustness to other type of noise as shown by (Ghosh et al. 2017) . Indeed, noise tolerance under simple non uniform noise and class conditional noise are also obtained but under some more assumptions (the true risk for the optimal classifier must be 0). Also, if we do not know the noise model, we would have to learn it. This can be more costly than using a model free method like a symmetric loss function.
>
> It is also possible to extend these ideas to regression by replacing the summation with an integral and choosing for example a normal distribution or a uniform distribution over a bounded interval. There is also a unique decomposition of any regression loss function as a sum of a term independent from the response variable y and a symmetric regression loss function.
>
> -Multi-class unhinged: We do not claim that the Multi-class unhinged would perform best in practice. We obtained better results with SGCE for example. However, it plays a central role in the theory of symmetric loss functions. Any symmetric loss function with strong enough regularization is approximately equivalent to the multi-class unhinged.

---

### Official Review · Reviewer_7oFd · 2023-11-09

**Soundness:** 3 good
**Presentation:** 4 excellent
**Contribution:** 3 good
**Rating:** 6
**Confidence:** 4

**Summary:**

This paper proposes a novel symmetrization method for multi-class loss functions, leading to a general approach for constructing symmetric loss functions from non-symmetric ones. The authors demonstrate the effectiveness of their method by applying it to various loss functions, including cross-entropy loss, generalized cross-entropy loss, and multi-class unhinged loss. Additionally, they provide theoretical insights into the properties of the multi-class unhinged loss function, showing that it is the unique convex, non-trivial, non-increasing, multi-class symmetric loss function under the assumption of invariance to permutations. Experimental results on the CIFAR10 dataset validate the robustness of the proposed approach.

**Strengths:**

The paper proposes a novel symmetrization method for multi-class loss functions, which is original in both its concept and its implementation. Also, the paper is also well-written and easy to understand. Authors present a variety of experimental results that validate the effectiveness of the proposed approach.

**Weaknesses:**

1. The paper presents experimental results on the CIFAR10 dataset, which is a relatively small dataset with simple label noise distributions. It would be more convincing to see results on a wider range of datasets, such as CIFAR100.

2. The comparison of the proposed symmetrization method to other existing methods for constructing symmetric loss functions is not clear. For example, Table 1 is not well structured. It is hard to find which result is the original loss, which are existing methods, and which is loss after symmetrization.

3. The proposed symmetrization method is only applicable to uniform label noise distributions while recent research is more focused on structured noise. Is there any approach to mitigate this issue such as using a different type of prior distribution?

**Questions:**

See weakness